

# Clinical significance of pretreatment prognostic nutritional index and lymphocyte-to-monocyte ratio in patients with advanced p16-negative oropharyngeal cancer—a retrospective study

Ming-Hsien Tsai[1,2,3], Tai-Lin Huang[2,4], Hui-Ching Chuang[1,2], Yu-Tsai Lin[1,2,3], Fu-Min Fang[2,5], Hui Lu[1] and Chih-Yen Chien[1,2,6]

[1] Department of Otolaryngology, Kaohsiung Chang Gung Memorial Hospital and Chang Gung University College of Medicine, Kaohsiung, Taiwan
[2] Kaohsiung Chang Gung Head and Neck Oncology Group, Cancer Center, Kaohsiung Chang Gung Memorial Hospital, Kaohsiung, Taiwan
[3] College of Pharmacy and Health Care, Tajen University, Pingtung County, Taiwan
[4] Department of Hematology and Oncology, Kaohsiung Chang Gung Memorial Hospital and Chang Gung University College of Medicine, Kaohsiung, Taiwan
[5] Department of Radiation Oncology, Kaohsiung Chang Gung Memorial Hospital, Chang Gung University College of Medicine, Kaohsiung, Taiwan
[6] Institute For Translational Research In Biomedicine, Kaohsiung Chang Gung Memorial Hospital, Kaohsiung, Taiwan

Corresponding author
Chih-Yen Chien,
cychien3965@adm.cgmh.org.tw

## ABSTRACT

**Background:** Systemic inflammation and nutritional status both play roles in the survival of cancer patients. Therefore, it is important to understand the effects of prognostic nutritional index (PNI) and lymphocyte-to-monocyte ratio (LMR) on the survival of patients with advanced p16-negative oropharyngeal cancer.

**Methods:** A total of 142 patients diagnosed with advanced p16-negative oropharyngeal cancer between 2008 and 2015 were enrolled in this study. All patients received primary treatment with definite concurrent chemoradiotherapy (CCRT). Optimal cutoff values for PNI and LMR were determined using receiver operating characteristic curves for survival prediction. Survival rates for different level of PNI and LMR were estimated and compared using Kaplan–Meier method and log-rank test to see if there were significant effects on these end points, including 5-year overall survival (OS), disease-specific survival (DSS) and disease-free survival (DFS) rates. The effects of PNI and LMR on survival were assessed using Cox regression model adjusted for other prognostic factors.

**Results:** The results showed the optimal cutoff values for PNI and LMR were 50.5 and 4.45, respectively. A high PNI (≧50.5) was significantly improved the 5-year OS. A low LMR (<4.45) was significantly associated with a poor 5-year DFS, DSS, and OS. In multivariate analysis, both PNI and LMR were independent prognosticators for 5-year OS.

**Conclusions:** Elevated pretreatment PNI and LMR are both favorable prognosticators in advanced p16-negative oropharyngeal cancer patients undergoing CCRT.

## INTRODUCTION

It is estimated that head and neck cancers are the sixth most commonly diagnosed systemic malignant tumors with more than 500,000 new cases and 300,000 associated deaths annually (*McGuire, 2016*). Oropharyngeal squamous cell carcinoma (OPSCC) is an aggressive type of head and neck cancer. The average annual percentage increase of OPSCC was 6.1% between 1980 and 2014, and the trend continues to increase with numerous OPSCC cases diagnosed as advanced stages in Taiwan (*Hsu et al., 2017*). Though investigations of OPSCC have been carried out for decades worldwide, its etiologic and clinical characteristics differ substantially among populations. For instance, human papillomavirus (HPV) infection may lead to a more favorable prognosis in patients than that without HPV infection. (*Mehanna et al., 2013*) Besides, the relatively low HPV infection rate, approximately 20–30%, in Taiwan along with the high prevalence of betel nut chewing both may deteriorate the prognosis, as investigated in our previous study (*Al-Swiahb et al., 2010*) makes the evaluation of treatments and possible prognostic factors in advanced stage HPV-negative OPSCC an urgent need.

In addition to HPV status, inflammatory biomarkers are thought to be a representation of the interaction between the tumor microenvironment and host immune system (*O'Callaghan et al., 2010*; *Aggarwal, Vijayalekshmi & Sung, 2009*). Recent studies have shown a negative prognostic value of higher neutrophil-to-lymphocyte ratio and lower lymphocyte-to-monocyte ratio (LMR) among patients with head and neck cancer (*Perisanidis et al., 2013*; *Haddad et al., 2015*; *Rassouli et al., 2015*; *Tham et al., 2018*). *Takahashi et al. (2019)* reported that a low LMR was an independent adverse prognostic factor for survival in patients with OPSCC.

Nutritional impairment has also been shown to have a negative impact on clinical outcomes (*Moon et al., 2016*). Patients with advanced stage OPSCC are often vulnerable to malnutrition at the time of diagnosis because of poor food intake due to cancer-related pain, mechanical obstruction by the tumor, or psychological problems. The prognostic nutritional index (PNI), calculated as previously described, (*Onodera, Goseki & Kosaki, 1984*) may be especially useful because it could act as a surrogate marker for both inflammation and nutritional status. This index was originally studied to demonstrate the correlation between postoperative complications and prognosis in patients with esophageal carcinoma (*Nozoe et al., 2002*). With regard to head and neck cancer, a low PNI had shown as a poor survival predictor in previous study (*Bruixola et al., 2018*).

Currently, studies on the role of PNI and LMR played in patients with advanced stage HPV-negative OPSCC are still limited. Clinically, p16 expression could be regarded as a surrogate marker for HPV in the prediction of tumor behavior in oropharyngeal cancer (*Golusiński et al., 2017*). Thus, the objective of this study was to identify the significant effects of PNI and LMR on clinical prognosis in patients with advanced stage (stage III/IV) p16-negative OPSCC.

## MATERIALS AND METHODS

### Study population

Patients who were histologically confirmed by biopsy to have stage III/IV p16-negative OPSCC were evaluated in the study. The TNM stage was reclassified according to the 8th edition of the American Joint Committee on Cancer (AJCC) staging system. Patients who were treated with primary concurrent chemoradiotherapy (CCRT) were eligible for this study. The determination of p16 expression in tumor cells by immunohistochemistry was done as suggested in the 8th edition AJCC staging system manual (*Amin et al., 2017*). Patients with clinical evidence of an acute infection within 4 weeks prior to the blood tests or who were diagnosed with recurrent tumors, distant metastases, other concomitant active cancers, or chronic inflammatory disease or who had a history of malignancy in the past 5 years were excluded from the study.

In this retrospective study, 142 patients with stage III/IV p16-negative OPSCC who underwent primary CCRT at the Kaohsiung Chang Gung Memorial Hospital in Taiwan between January 2008 and April 2015 were recruited. Treatment was primarily based on the American NCCN guidelines. The chemotherapy agent was cisplatin-based and the radiation technique for all patients was intensity-modulated radiation therapy (IMRT). The primary radiation dose for all of our patients was between 70 and 74 Gy with conventional fractionated daily dose of 1.8 or 2 Gy. All included patients completed the treatment programs formulated by the multidisciplinary team.

### Variables and outcomes

Pretreatment clinical variables of interest were collected, including age, sex, Adult Comorbidity Evaluation-27 (ACE-27) score and clinical TNM stage of the tumor. Information collection of pretreatment complete blood count (including absolute lymphocyte and monocyte counts) and biochemical (including albumin) tests using the peripheral blood sample were also conducted within one week before treatment.

The LMR was calculated by dividing the baseline absolute peripheral lymphocyte count (cells/mm$^3$) by the absolute monocyte peripheral count (cells/mm$^3$).

The PNI was calculated as follows: 10 × baseline serum albumin (g/dL) + 0.005 × baseline absolute lymphocyte count (cells/mm$^3$).

### Statistical analysis

The endpoints in our study were the 5-year overall survival (OS), 5-year disease-specific survival (DSS), and 5-year disease-free survival (DFS) rates. OS calculated the time frame from the date of the first treatment to the date of death or last follow-up; disease specific survival calculated from the date of the first treatment to the date of death because of tumor or last follow-up. Disease free survival calculated the time from the date of the first treatment to the date of recurrence, metastasis, or last follow-up. Follow-up was continued through May 2020. Receiver operating characteristic curves for survival were plotted, and Youden's index, which calculated as J = sensitivity + specificity − 1, was applied to verify the optimum cutoff value of LMR and PNI for OS. Survival rates of certain prognostic factors were estimated using the Kaplan–Meier method, and the log-rank test

was used to determine the heterogeneity of each specific factor. Sex and smoking status variables were excluded from the analysis because of the extremely imbalanced distribution. The Cox proportional hazards model was built with independent primary factors and other significant prognostic factors that were identified in prior univariate survival analyses. The variance inflation factors (VIF) were assessed to avoid multicollinearity among independent variables in the Cox model. Both VIF values for continuous PNI to continuous LMR or dichotomous PNI to dichotomous LMR were below 3 (1.004 and 1.002, respectively), which indicated that there was a low correlation between PNI and LMR. All statistics tests were two-sided, with 0.05 significant level. All statistical analyses were performed using the Social Science Software, version 20.0 package (SPSS, Chicago, IL, USA). This study was approved by the Medical Ethics and Human Clinical Trial Committees at Chang Gung Memorial Hospital (Ethical Application Reference number: 202000471B0). Patients' consent to review their medical records was not required by this hospital's committees because the patient data remained anonymous in this study.

# RESULTS

Of the 142 p16-negative OPSCC patients, 99.3% (141) were men and 0.7% (1) were women. The mean age at diagnosis for all participants was 53.8 years (range: 36–85 years). The mean follow-up period was 40.7 months (3.6–111.8 months). Nine patients (6.3%) had stage III disease, 34 (23.9%) had stage IVA and 99 (69.7%) had stage IVB. This cohort included patients with clinical T classifications of T1 ($n = 4$, 2.8%), T2 ($n = 24$, 16.9%), T3 ($n = 24$, 16.9%), T4a ($n = 34$, 23.9%) and T4b ($n = 56$, 39.4%). Clinical nodal metastasis was present in 117 patients (82.4%) and 65 (45.8 %) had extranodal extensions (ENE). The clinicopathological features of the 142 cases, and their survival outcomes were listed in Table 1.

The optimal cutoff value for PNI was 50.5, and 4.45 for LMR (Fig. 1). Patients with PNI $\geq$ 50.5 or LMR $\geq$ 4.45 did not have significant correlations with age, T classification, N classification, or other clinicopathologic factors (all $p > 0.05$; Table S1).

The OS rate for patients with PNI $\geq$ 50.5 was significantly higher than that for patients with PNI < 50.5 (48.1% vs. 24.7%, $p = 0.004$). Similarly, the DSS for patients with PNI $\geq$ 50.5 was significantly higher than that for patients with PNI < 50.5 (57.2% vs. 42%, $p = 0.043$). Moreover, DFS had a similar trend by PNI difference in our cohort (44.3% vs. 34.2%), although $p$ value did not reach statistical significance ($p = 0.108$, Fig. 2). Regarding the LMR, the 5-year OS (55.5% vs. 26.6%), DSS (66.8% vs. 41.4%) and DFS (51.4% vs. 35.0%) were all significantly increased (both $p < 0.05$, Fig. 3) among patients with LMR $\geq$ 4.45, compared with those with LMR < 4.45. Clinically positive ENE status was another significant predictor of poor outcome for 5-year OS, DSS and DFS in univariate analysis (Table 2).

Multiple regression analysis was applied to analyze the relationship between survival outcome and significant factors which were revealed in prior univariate analyses. PNI was an independent factor of OS in this cohort (hazard ratio (HR): 1.778, 95% confidence

**Table 1 Clinical characteristics of 142 patients who were diagnosed with advanced p16-negative OPSCC.**

| Characteristics | | Value | % |
|---|---|---|---|
| Mean age (range), year | | 53.8 (36, 85) | |
| Mean follow up time (range), months | | 40.7 (3.7, 111.8) | |
| Sex | Male | 141 | 99.3 |
| | Female | 1 | 0.7 |
| Clinical TNM Stage | III | 9 | 6.3 |
| | IVA | 34 | 23.9 |
| | IVB | 99 | 69.7 |
| Clinical T classification | T1 | 4 | 2.8 |
| | T2 | 24 | 16.9 |
| | T3 | 24 | 16.9 |
| | T4a | 34 | 23.9 |
| | T4b | 56 | 39.4 |
| Clinical N classification | N0 | 25 | 17.6 |
| | N1 | 15 | 10.6 |
| | N2b | 17 | 12 |
| | N2c | 20 | 14.1 |
| | N3b | 65 | 45.8 |
| Clinical ENE | Negative | 77 | 54.2 |
| | Positive | 65 | 45.8 |
| PNI | Unknown | 7 | 4.9 |
| | <50.5 | 79 | 55.6 |
| | ≥50.5 | 56 | 39.4 |
| LMR | <4.45 | 99 | 69.7 |
| | ≥4.45 | 43 | 30.3 |
| Recurrence | No | 59 | 41.5 |
| | Yes | 83 | 58.5 |
| Last status | NED | 35 | 24.6 |
| | Alive with disease | 12 | 8.5 |
| | DOD | 68 | 47.9 |
| | DWOD | 27 | 19.0 |

**Note:**
OPSCC, oropharyngeal squamous cell carcinoma; PNI, prognostic nutritional index = 10 × serum albumin (g/dl) + 0.005 × total lymphocyte count (/mm$^3$); ENE, extranodal extension; LMR, lymphocyte to monocyte ratio; NED, no evidence of disease; DOD, died of disease; DWOD, die without disease.

interval (CI) [1.145–2.761]) and simultaneously adjusted by other independent factors, LMR and ENE (Table 3). In another model, the status of LMR showed a significant prognosticator in OS (HR of 2.408, 95% CI [1.439–4.029]), DSS (HR: 2.33, 95% CI [1.255–4.323]) and DFS (HR: 1.765, 95% CI [1.067–2.892]) after being adjusted by other factors (Tables 3–5). The status of clinical ENE was a significant prognosticator of OS (HR: 1.592, 95% CI [1.054–2.405]), DSS (HR: 2.159, 95% CI [1.319–3.533]) and DFS (HR: 1.86, 95% CI [1.202–2.878]) (Tables 3–5).

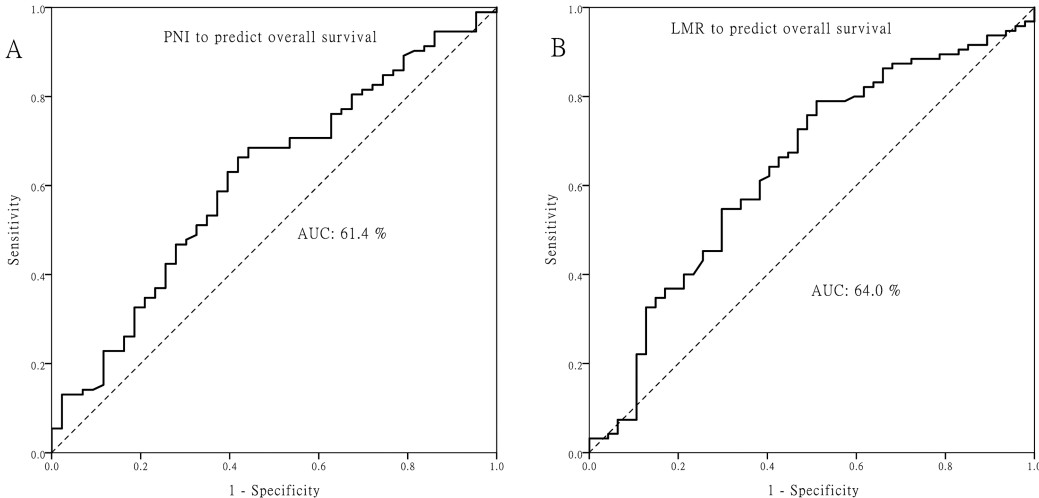

**Figure 1 Receiver operating characteristic curves.** Receiver operating characteristic curves for predicting the survival outcome. (A) Pretreatment prognostic nutritional index (PNI). (B) Pretreatment lymphocyte to monocyte ratio (LMR).

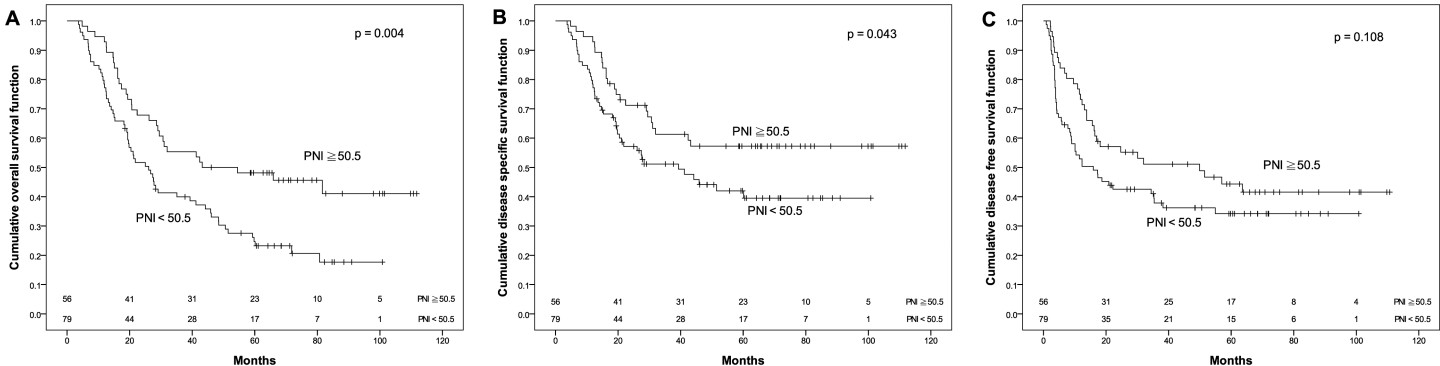

**Figure 2 Kaplan–Meier survival curves.** Kaplan–Meier survival curves by different level of pretreatment prognostic nutritional index (PNI). (A) Overall survival. (B) Disease-specific survival. (C) Disease-free survival.

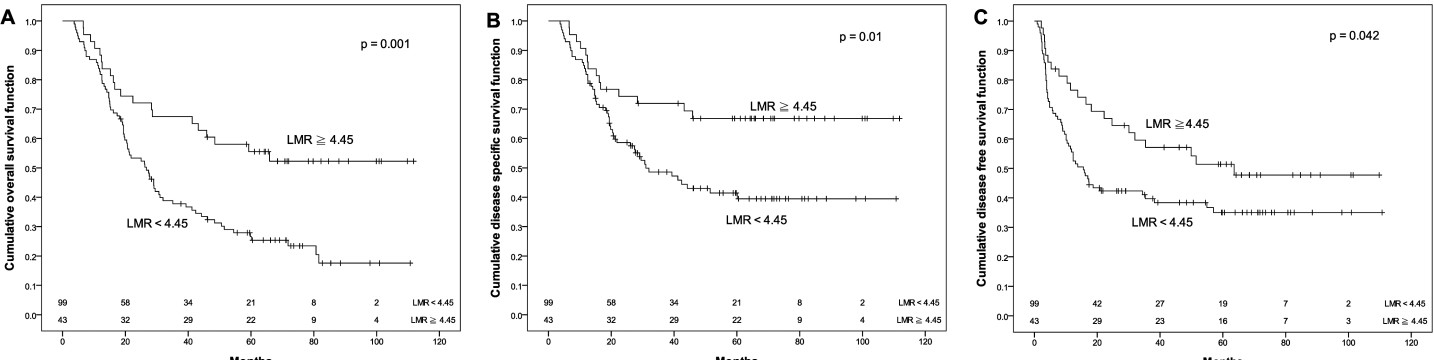

**Figure 3 Kaplan–Meier survival curves.** Kaplan–Meier survival curves by different level of pretreatment lymphocyte to monocyte ratio (LMR). (A) Overall survival. (B) Disease-specific survival. (C) Disease-free survival.

**Table 2 Univariate analysis of factors impacting survival ($n = 142$).**

| Variable | | Number | Event | 5 year OS (%) | $p$ | Event | 5 year DSS (%) | $p$ | Event | 5 year DFS (%) | $p$ |
|---|---|---|---|---|---|---|---|---|---|---|---|
| Age | <53 | 70 | 45 | 38.1 | 0.489 | 34 | 49.5 | 0.836 | 44 | 35.3 | 0.743 |
| | ≧53 | 72 | 50 | 33.2 | | 34 | 50.1 | | 39 | 44.3 | |
| ACE-27 | 0 | 93 | 60 | 37.9 | | 44 | 50.4 | | 56 | 39.1 | |
| | 1 | 42 | 30 | 31.4 | 0.602 | 21 | 47.7 | 0.948 | 24 | 39.9 | 0.893 |
| | 2 | 7 | 5 | 28.6 | | 3 | 57.1 | | 3 | 57.1 | |
| Betel nut chewing | No | 36 | 26 | 36.1 | 0.841 | 16 | 55.7 | 0.672 | 18 | 48.7 | 0.416 |
| | Yes | 106 | 69 | 35.5 | | 52 | 47.7 | | 65 | 36.9 | |
| Alcohol drinking | No | 24 | 13 | 49.0 | 0.226 | 9 | 61.6 | 0.327 | 11 | 52.1 | 0.221 |
| | Yes | 118 | 82 | 32.8 | | 59 | 46.9 | | 72 | 37.3 | |
| Clinical T classification | T1/2/3 | 52 | 34 | 33.5 | 0.693 | 24 | 50.5 | 0.638 | 29 | 43.6 | 0.407 |
| | T4a/b | 90 | 61 | 36.7 | | 44 | 49.4 | | 54 | 37.7 | |
| Clinical N classification | N0 | 25 | 18 | 35.6 | 0.684 | 11 | 50.7 | 0.801 | 15 | 36.3 | 0.884 |
| | N1–N3b | 117 | 77 | 35.5 | | 57 | 49.6 | | 68 | 40.6 | |
| Clinical ENE | Negative | 77 | 48 | 41.0 | 0.037* | 29 | 59.4 | 0.003* | 38 | 49.7 | 0.008* |
| | Positive | 65 | 47 | 29.1 | | 39 | 38.6 | | 45 | 28.2 | |
| PNI | <50.5 | 79 | 61 | 24.7 | 0.004* | 43 | 42.0 | 0.043* | 50 | 34.2 | 0.108 |
| | ≧50.5 | 56 | 31 | 48.1 | | 23 | 57.2 | | 31 | 44.3 | |
| LMR | <4.45 | 99 | 75 | 26.6 | 0.001* | 54 | 41.4 | 0.01* | 62 | 35.0 | 0.042* |
| | ≧4.45 | 43 | 20 | 55.5 | | 14 | 66.8 | | 21 | 51.4 | |

Notes:
* Statistically significant ($p < 0.05$).
OS, overall survival; DSS, disease specific survival; DFS, disease free survival; ACE-27, adult comorbidity evaluation-27; ENE, extranodal extension; PNI, prognostic nutritional index; LMR, lymphocyte to monocyte ratio.

**Table 3 Multivariate analysis of prognostic factors associated to overall survival.**

| Factor | Hazard ratio | 95% CI | $p$-Value |
|---|---|---|---|
| PNI | | | 0.01* |
| ≧50.5 | 1 | | |
| <50.5 | 1.778 | [1.145–2.761] | |
| LMR | | | 0.001* |
| ≧4.45 | 1 | | |
| <4.45 | 2.408 | [1.439–4.029] | |
| ENE | | | 0.027* |
| Negative | 1 | | |
| Positive | 1.592 | [1.054–2.405] | |

Notes:
* Statistically significant ($p < 0.05$).
ENE, extranodal extension; PNI, prognostic nutritional index; LMR, lymphocyte to monocyte ratio.

## DISCUSSION

In the current study of patients with advanced stage (stage III/IV) p16-negative OPSCC, the 5-year DFS, DSS, and OS rates were 39.9%, 49.8% and 35.6%, respectively. PNI, LMR and clinical ENE status were all significant independent factors of OS in our multivariate cox regression analysis.

**Table 4  Multivariate analysis of prognostic factors associated to disease-specific survival.**

| Factor | Hazard ratio | 95% CI | p-Value |
|---|---|---|---|
| PNI | | | 0.066 |
| ≧50.5 | 1 | | |
| <50.5 | 1.624 | [0.968–2.723] | |
| LMR | | | 0.007* |
| ≧4.45 | 1 | | |
| <4.45 | 2.33 | [1.255–4.323] | |
| ENE | | | 0.002* |
| Negative | | | |
| Positive | 2.159 | [1.319–3.533] | |

Notes:
* Statistically significant (p < 0.05).
ENE, extranodal extension; PNI, prognostic nutritional index; LMR, lymphocyte to monocyte ratio.

**Table 5  Multivariate analysis of prognostic factors associated to disease-free survival.**

| Factor | Hazard ratio | 95% CI | p-Value |
|---|---|---|---|
| LMR | | | 0.027* |
| ≧4.45 | 1 | | |
| <4.45 | 1.765 | [1.067–2.892] | |
| ENE | | | 0.005* |
| Negative | 1 | | |
| Positive | 1.86 | [1.202–2.878] | |

Notes:
* Statistically significant (p < 0.05).
ENE, extranodal extension; LMR, lymphocyte to monocyte ratio.

Clinical ENE is the extension of metastatic lymph node through an affected lymph node capsule. It has always been considered a marker of poor prognosis as tumor recurrence and oncological survival in head and neck cancer; thus, it was proposed to be incorporated into the newest edition of the AJCC staging system manual (*Amin et al., 2017*). Our cohort also revealed similar results, showing that the presence of ENE was associated with poor oncologic outcomes.

A low PNI indicated a decrease in the serum albumin and/or a low absolute lymphocyte count. Serum albumin is an important factor of the host inflammatory response and nutritional status (*Gupta & Lis, 2010*). The absolute lymphocyte count is also believed to be an important participant in the inhibition of cancer growth by initiating a cytotoxic immune response (*Mantovani et al., 2008*). Taken together, this existing evidence showed that malnutrition and lymphocytopenia may be factors affecting a chronically impaired immune system. The cutoff value for PNI reported in previous studies in other type of cancer was 40–60 (*Feng & Chen, 2014*; *Lee et al., 2017*; *Jian-Hui et al., 2016*; *Shibutani et al., 2015*; *Yang et al., 2016*; *Zhang et al., 2019*). With regard to head and neck cancer, several studies found that lower PNI predicted poor oncologic outcomes in head and neck squamous cell carcinoma (HNSCC) (Table 6) (*Bruixola et al., 2018*;

**Table 6 Different studies about PNI in HNSCC.**

| Reference | Site | Case number | Cut off for PNI | Primary treatment strategy | Statically significant Outcome measurement |
|---|---|---|---|---|---|
| *Bruixola et al. (2018)* | Locoregionally advanced HNSCC | 145 | 45 | ICT followed by CCRT | OS |
| *Kono et al. (2017)* | HNSCC | 101 | 40 | Radiotherapy | Toxicity of radiotherapy |
| *Chang et al. (2018)* | Advanced oral cavity, oropharynx, hypopharyngeal cancer | 143 | 36 | CCRT | Treatment tolerance and toxicity of CCRT |
| *Fu et al. (2016)* | Laryngeal squamous cell carcinoma | 975 | 48.65 | Radical surgery | DSS and OS |
| Our current study | Advanced stage p16 negative OPSCC | 142 | 50.5 | CCRT | OS |

**Note:**
PNI, prognostic nutritional index; HNSCC, head and neck squamous cell carcinoma; OPSCC, oropharyngeal squamous cell carcinoma; ICT, induction chemotherapy; CCRT, concurrent chemoradiotherapy; OS, overall survival; DSS, disease-specific survival.

**Table 7 Different studies about LMR in HNSCC.**

| Reference | Site | Case number | Cut off for LMR | Primary treatment strategy | Statically significant Outcome measurement |
|---|---|---|---|---|---|
| *Takahashi et al. (2019)* | Oropharyngeal carcinoma | 75 | 4.97 | Heterogeneity (76% of population were CRT) | OS |
| *Tham et al. (2019)* | HNSCC | 123 | 2.8 | Radical surgery | Event free survival |
| *Furukawa et al. (2019)* | Tongue cancer | 103 | 4.29 | Radical surgery | OS |
| *Yang et al. (2018)* | Hypopharyngeal carcinoma | 197 | 2.98 | Not well documented | OS, DSS and DFS |
| *Kano et al. (2017)* | Oropharyngeal, hypopharyngeal, and laryngeal cancers | 285 | 3.22 | Concurrent CRT | OS and DFS |
| Our current study | Advanced stage p16 negative OPSCC | 142 | 4.45 | Concurrent CRT | OS, DSS and DFS |

**Note:**
LMR, lymphocyte to monocyte ratio; HNSCC, head and neck squamous cell carcinoma; OPSCC, oropharyngeal squamous cell carcinoma; CRT, chemoradiotherapy; OS, overall survival; DSS, disease-specific survival; DFS, disease-free survival.

*Kono et al., 2017*; *Chang et al., 2018*; *Fu et al., 2016*). *Bruixola et al. (2018)* demonstrated that low PNI (cutoff value: 45) was an independent prognostic biomarker in locoregional advanced HNSCC. *Fu et al. (2016)* studied 975 patients with laryngeal squamous cell carcinoma treated with curative laryngectomy, and found that patients with PNI < 48.65 had a low probability of cancer-specific survival and OS. Our results are comparable with these findings, showing that a low PNI is an indicator of poor prognosis in patients with advanced stage (stage III/IV) p16-negative OPSCC undergoing primary CCRT, with a cutoff value similar to previous studies (*Bruixola et al., 2018*; *Kono et al., 2017*; *Chang et al., 2018*; *Fu et al., 2016*). In our study, patients with PNI < 50.5 had significantly reduced survival with adjusted for other prognostic factors in the multivariate analysis.

Studies of investigating the clinical effects of LMR on HNSCC prognosis have increased in recent years. White blood cell differential could be divided into myeloid lineage and lymphoid lineage. It is believed that lymphoid lineage preponderance of white blood cell was related to better survival based on previous HNSCC study (*Wu et al., 2017*). Several studies found that lower LMR predicted reduced DSS and OS in HNSCC (Table 7) (*Takahashi et al., 2019*; *Tham et al., 2019*; *Furukawa et al., 2019*; *Yang et al., 2018*;

*Kano et al., 2017*). In addition, the relationship between LMR and advanced stage OPSCC was not thoroughly evaluated. Our results are comparable with these findings, showing that a low LMR is an indicator of poor prognosis in advanced stage (stage III/IV) p16-negative OPSCC. Patients with LMR < 4.45 have significantly reduced OS, DSS and DFS according to the multivariate analysis.

The mechanism between an increased systemic inflammatory response and promotion of tumor cell invasion is not clearly understood. A possible explanation might lie in the antitumoral roles that lymphocyte plays by inhibiting tumor cell proliferation and migration, and reinforcing human's immune response to cancer (*De Giorgi et al., 2012*). Fewer infiltrating lymphocytes have been correlated to poor prognosis (*Gooden et al., 2011*). In contrast, higher levels of monocyte-derived macrophages have been associated with greater tumor aggressiveness and poorer survival outcomes (*Pollard, 2004*). This is postulated to happen through tumor microenvironment mediators such as TNF-α, vascular endothelial growth factor and epidermal growth factor (*Pollard, 2004*; *Xiong et al., 1998*). A low LMR implies a relative decrease in lymphocytes and/or increase monocytes. Perhaps, the prognostic ability of LMR is owing to its action as a crude marker for the pro-tumor versus anti-tumor dynamic in the immune system (*Lin, Chien & Chuang, 2017*).

PNI, which calculated as $10 \times$ baseline serum albumin (g/dL) + $0.005 \times$ baseline absolute lymphocyte count (cells/mm$^3$), is used to evaluate the immune-nutritional status and may influence the prognosis of cancer patients (*Yao et al., 2013*). Poor immune-nutritional status has been reported as its' association with an immunosuppressed condition, which provides a favorable microenvironment for tumor relapse (*Colotta et al., 2009*). That may be the reason why this immunosuppressed condition in low-PNI patients may cause the poor outcomes. Recently, remarkable progress in research on immune checkpoints in tumor immunity has allowed the elucidation of the molecular mechanism underlying immunological tolerance to tumor development. The association between peripheral inflammatory biomarkers and treatment outcomes for immunotherapy remains unclear. These biomarkers might serve as a useful predictor for immunotherapy in the treatment of head and neck cancer in the future.

In our study, we have identified the clinical significance of PNI and LMR on survival in patients with p16 negative oropharyngeal cancer treated by CCRT. Moreover, we had control cancer stage and HPV status these two well-known prognostic factors in oropharyngeal cancer, making this cohort homogenous for our analysis findings. However, the drawback of our study is that it is retrospective, and selective bias may exist. A prospective study or large series study from multiple institutes is necessary to confirm our findings.

## CONCLUSION

In summary, our current study showed that patients with higher pretreatment LMR ($\geq$4.45) showed significantly better survival than those with lower LMR (<4.45); Patients with higher PNI ($\geq$50.5) revealed significantly better 5-year OS and 5-year DSS than those with lower PNI (<50.5). According to Cox regression analysis from this

cohort, pretreatment LMR and PNI were also an independent prognostic factor that predicts OS. Interestingly, it may be possible to incorporate pretreatment LMR and PNI into the treatment strategy for patients with advanced stage p16-negative OPSCC undergoing CRT/RTO in the future.

### Funding
This work was supported by grants nos. CMRPG 8D1421 and CORPG8F1481-3 from Kaohsiung Chang Gung Memorial Hospital. There was no additional external funding received for this study. The funders had no role in study design, data collection and analysis, decision to publish, or preparation of the manuscript.

### Grant Disclosures
The following grant information was disclosed by the authors:
Kaohsiung Chang Gung Memorial Hospital: CMRPG 8D1421 and CORPG8F1481-3.

### Competing Interests
The authors declare that they have no competing interests.

### Author Contributions
- Ming-Hsien Tsai conceived and designed the experiments, analyzed the data, prepared figures and/or tables, authored or reviewed drafts of the paper, and approved the final draft.
- Tai-Lin Huang performed the experiments, authored or reviewed drafts of the paper, and approved the final draft.
- Hui-Ching Chuang conceived and designed the experiments, authored or reviewed drafts of the paper, and approved the final draft.
- Yu-Tsai Lin conceived and designed the experiments, authored or reviewed drafts of the paper, and approved the final draft.
- Fu-Min Fang performed the experiments, authored or reviewed drafts of the paper, and approved the final draft.
- Hui Lu analyzed the data, prepared figures and/or tables, authored or reviewed drafts of the paper, and approved the final draft.
- Chih-Yen Chien conceived and designed the experiments, performed the experiments, analyzed the data, prepared figures and/or tables, authored or reviewed drafts of the paper, and approved the final draft.

### Human Ethics
The following information was supplied relating to ethical approvals (i.e., approving body and any reference numbers):
This study was approved by the Medical Ethics and Human Clinical Trial Committees at Chang Gung Memorial Hospital (Ethical Application Reference number:202000471B0).

## Data Availability

The raw measurements are available in the Supplementary Files.

## Supplemental Information

Supplemental information for this article can be found online at http://dx.doi.org/10.7717/peerj.10465#supplemental-information.

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
