# Peer review of "Clinical significance of pretreatment prognostic nutritional index and lymphocyte-to-monocyte ratio in patients with advanced p16-negative oropharyngeal cancer—a retrospective study"

_PeerJ, doi:10.7717/peerj.10465_

## Round 0.1 · original submission · Minor Revisions

Dear Dr. Chien,
Our reviewers have checked the manuscript and based on their recommendation your manuscript can be considered after minor revisions. While revising, in addition to providing the requested information and modifications, I would strongly recommend that the manuscript be professionally assessed for English grammar and syntax.

·

Basic reporting

The manuscript lacks syntax in English and grammar, such as lines 31, 34, 36, 53, 58 are not understandable. The authors are suggested to improve on it.

There are a few missing references, such as lines 82, 187, 193.

The manuscript carries all the required scientific information and is well structured.

The manuscript contains relevant results.

Experimental design

1) Authors are suggested to perform the correlation analysis of PNI and LMR with tumor recurrence & extra-nodal extension and make a graph using the Table 1 data. This correlation will be helpful in understanding the direct relationship of PNI and LMR with tumor growth and staging.

2) Similarly, the authors are suggested to make a graphical representation of PNI and LMR data correlation with other critical parameters such as tobacco use, drinking, alcohol consumption using the data from Table 2.

3) In Table 7, I see that different head and neck studies have proposed different LMR values which are ranging from 2.8 to 4.97. It would great if authors can comment on this variation, and what could be the factor affecting the LMR score? This information might be helpful in selecting the criteria along with LMR calculation.

4) Line 141 says that 66.9% of patients were dead at the very end of the study then how the PNI and LMR correlation values were calculated with OPSCC patients DFS and OS.

5) It would be nice if authors can add the details of “Youden’s Index” calculation and “variance inflation factor” in the materials and method section.

Validity of the findings

no comments

·

Basic reporting

The manuscript entitled “Clinical significance of pretreatment prognostic nutritional
index and lymphocyte-to-monocyte ratio in patients with advanced p16-negative oropharyngeal cancer” by Tsai et al. describes the effect of PNI and LMR on the survival of patients with advanced p16-negative oropharyngeal cancer. The study included 142 patients for the determination of cut off values of PNI and LMR for improved survival. The results show the higher survival rate in patients with higher levels of PNI and LMR. The manuscript is well written, and interesting. Discussion section is also very informative where authors have explained the possible mechanism for the effect of PNI and LMR on the survival rate. However, it is important to address few questions to further improve the manuscript.

Experimental design

Authors have considered two factors LMR and PNI. Did authors also look at other factors say platelet count or neutrophil or total WBC count? If authors include these values, it will complete the data of blood profiling and also will strengthen the manuscript which will help further research for oropharyngeal cancer treatment.

What are the post-treatment values of PNI and LMR? Authors have not mentioned about post-treatment conditions.

Validity of the findings

Did authors also collect data for the patients if they had any underlying condition which effect patients’ survival?

The importance of current study should be discussed in the discussion section. For example, how this study can be useful for future therapy of p16-negative oropharyngeal cancer.

Reviewer 3 ·

Basic reporting

The authors have utilized a number of clinical parameters and determined that pre-treatment nutritional index (PNI) and lymphocyte to monocyte ratio (LMR) is a potential prognostic indicator of overall survival in patients with advanced p16 negative oropharyngeal carcinoma. The study is generally well written and concise but can benefit from some additional details within the methods section and discussion.

Experimental design

Experimental design and set up is adequate for the questions the authors have attempted to answer in this study. However, there is lack of clarity on the different factors utilized for multivariate analyses in tables 3-5 (which also are not referenced in the text). The authors have chosen to focus on PNI and LMR as prognostic factors without discussing much about ENE as the only other significant predictor in univariate analysis. The figures are not as self explanatory and can benefit from modifications as suggested below-

1. Figure 1 A and B need clear annotations/ legends as to what they are depicting. It is unclear which curve represents PNI and LMR. The figures show an AUC of 61 and 64% respectively, which although not unacceptable, are on the lower side of what would be considered ideal for such analyses.
2. Figure 2 and 3 would benefit from adding the total number of patients for each curve as well as the hazard ratios in addition to the existing p values in the plot

Validity of the findings

The findings are consistent with previously reported literature and the authors have carefully discussed the possible explanations for their observations.

---

## Round 0.2 · Minor Revisions

Hi Dr. Chien,
Congratulations, our reviewers have conveyed their satisfaction of your rebuttal. However, before I can accept your manuscript I would request you to address the two minor suggestions pointed out by Reviewer 2 (see below)
* * *
Line 226, "That’s may be the reason." needs language change.

Line 233-238, “Several limitations should be addressed in current study”. This sentence itself is not clear. The entire paragraph should be changed to strengthen the manuscript. For instance, why the mentioned studies were not done, should be described in an effective way so that it does not undermine the current manuscript.
* * *
·

Basic reporting

The Authors have improved on syntax in English and grammar. The authors have made the necessary improvements and have cited the relevant literature references. Moreover, the manuscript is well structured and has all relevant data and figures to support the observation and investigation.

Experimental design

The experimental design is based on a previously published report which seems to be standard for PNI and LMR investigation in patient samples. Additionally, the data collection and analysis from 142 patient samples make the study more impactful.

Validity of the findings

The findings of this article are in sequence with the previously published report. In the revised manuscript, the authors have made necessary changes that further support the study.

·

Basic reporting

The revised submission of the manuscript entitled “Clinical significance of pretreatment prognostic nutritional index and lymphocyte-to-monocyte ratio in patients with advanced p16-negative oropharyngeal cancer – a retrospective study (#51116)”, describes the effect of nutritional status and inflammation on the survival of cancer patients. Authors have answered all the comments/concerns point-wise and improved the manuscript accordingly. The abstract, introduction and discussion sections are improved very well.

Line 226, "That’s may be the reason." needs language change.

Line 233-238, “Several limitations should be addressed in current study”. This sentence itself is not clear. The entire paragraph should be changed to strengthen the manuscript. For instance, why the mentioned studies were not done, should be described in an effective way so that it does not undermine the current manuscript.

Experimental design

no comment

Validity of the findings

no comment

Reviewer 3 ·

Basic reporting

Article meets standards

Experimental design

No comment

Validity of the findings

Authors have made necessary changes. No further changes required

---

## Round 0.3 · accepted · Accept

Dear Dr. Chien,
Congratulations, your manuscript has been now accepted for publication.